# MHC Class I Downregulation in Cancer: Underlying Mechanisms and Potential Targets for Cancer Immunotherapy

**DOI:** 10.3390/cancers12071760

**Published:** 2020-07-02

**Authors:** Annelisa M. Cornel, Iris L. Mimpen, Stefan Nierkens

**Affiliations:** 1Center for Translational Immunology, University Medical Center Utrecht, Utrecht University, 3584 CX Utrecht, The Netherlands; a.m.cornel@umcutrecht.nl (A.M.C.); mimpeniris@gmail.com (I.L.M.); 2Princess Máxima Center for Pediatric Oncology, Utrecht University, 3584 CS Utrecht, The Netherlands

**Keywords:** MHC-I downregulation, cancer immunotherapy, antigen presentation, tumor immunogenicity, adaptive immune involvement

## Abstract

In recent years, major advances have been made in cancer immunotherapy. This has led to significant improvement in prognosis of cancer patients, especially in the hematological setting. Nonetheless, translation of these successes to solid tumors was found difficult. One major mechanism through which solid tumors can avoid anti-tumor immunity is the downregulation of major histocompatibility complex class I (MHC-I), which causes reduced recognition by- and cytotoxicity of CD8^+^ T-cells. Downregulation of MHC-I has been described in 40–90% of human tumors, often correlating with worse prognosis. Epigenetic and (post-)transcriptional dysregulations relevant in the stabilization of NFkB, IRFs, and NLRC5 are often responsible for MHC-I downregulation in cancer. The intrinsic reversible nature of these dysregulations provides an opportunity to restore MHC-I expression and facilitate adaptive anti-tumor immunity. In this review, we provide an overview of the mechanisms underlying reversible MHC-I downregulation and describe potential strategies to counteract this reduction in MHC-I antigen presentation in cancer.

## 1. Introduction

In recent years, major advances have been made in cancer immunotherapy, thereby drastically improving the prognosis of cancer patients. Several types of Food and Drug Administration (FDA)-approved immunotherapies, such as checkpoint inhibitors (CPI), chimeric antigen receptor (CAR) T-cells, and dendritic cell vaccines, aim to boost T-cell-mediated cytotoxicity to combat cancer. These treatments led to increased survival chances, particularly for patients suffering from hematological cancers, but translation to the solid tumor setting was found difficult. Additional immune escape mechanisms, including immune checkpoint expression, induction of immunosuppressive immune subsets (e.g., regulatory T-cells and myeloid-derived suppressor cells (MDSCs)), loss of immunogenic antigens, and decreased antigen presentation allow these tumors to evade anti-tumor immunity (reviewed by Sharma et al. [1]), posing serious challenges to overcome in order to improve therapy response.

One way in which tumors can avoid tumor-associated antigen presentation, and therewith T-cell-mediated cytotoxicity, is the downregulation of surface display of major histocompatibility complex (MHC) class I, a crucial factor in the initiation of an adaptive immune response. The importance of MHC-I downregulation in immune evasion is substantiated by the observed correlations between MHC-I expression on tumor cells and the amount of tumor infiltrating lymphocytes (TILs) in several cancers [2,3]. Furthermore, several groups have reported impaired MHC-I antigen processing and presentation as a predictor of (acquired) resistance to CPI therapy [4,5,6,7,8] and adoptive cell therapy [9,10,11].

Downregulation of MHC-I has been described in 40–90% of human tumors [9,12,13,14,15,16,17,18,19,20], often correlating with worse prognosis [18,21,22,23,24,25,26,27,28,29]. Both adult and pediatric tumors are able to reduce MHC-I surface display by the use of different regulatory mechanisms. Where adult tumors downregulate MHC-I expression in order to escape from the immune system, pediatric cancers, such as neuroblastoma, often arise from embryonic tissues that intrinsically lack immunological features, potentially explaining the low expression of MHC-I in these cancer types [30].

The intrinsic reversible nature of most MHC-I dysregulations provides an opportunity to restore MHC-I antigen presentation and facilitate adaptive anti-tumor immunity. This review aims to provide an overview of the mechanisms underlying reversible MHC-I downregulation and demonstrate potential therapeutic targets to induce MHC-I expression and improve T-cell-mediated cytotoxicity in cancer.

## 2. Dysregulation of MHC-I Expression in Cancer

T-cell receptors (TCRs) of CD8^+^ T-cells can only bind to their targets in the context of MHC-I, which is expressed on all nucleated cells. MHC-I presents endogenous antigens, a process important for reporting intracellular changes, for example caused by viral infections or malignant transformation, to the immune system in order to initiate a CD8^+^ T-cell response. The heterodimer MHC-I consists of a heavy chain, encoded by the human leukocyte antigen (HLA)-A, HLA-B, and HLA-C genes, and an invariant light chain called β_2_-microglobulin (β_2_M). The heterodimer requires stabilization by a peptide, which is loaded into the MHC-I peptide-binding groove by the antigen processing machinery (APM).

Antigen presentation in MHC-I context is a complex, multi-step process which can be dysregulated at many levels. MHC-I and β_2_M are synthesized in the endoplasmic reticulum (ER) and require stabilization by chaperone proteins (e.g., calreticulin, ERp57, and tapasin). Designated intracellular proteins are targeted for degradation by ubiquitination, after which they undergo proteasomal degradation into peptides. These peptides are subsequently translocated into the ER by the transporter associated with antigen processing (TAP) and bind to MHC-I directly or after further processing by ER aminopeptidases (ERAP1 and ERAP2). Interaction between tapasin and TAP allows translocation of peptides into the MHC-I binding groove, release of chaperone proteins, and stabilization of the MHC-I complex. Finally, the MHC-I-antigen complex travels via the Golgi apparatus to the cell surface, where antigen presentation to CD8^+^ T-cells can start. In cancer, one or several proteins in this complex pathway can be dysregulated, which may have major consequences on cell surface display of MHC-I (Figure 1).

Although downregulation of surface expression of MHC-I allows evasion from T-cell-mediated anti-tumor immunity, low MHC-I expression sensitizes cells to Natural Killer (NK) cell-mediated cytotoxicity. MHC-I functions as an inhibitory ligand for NK-cells by binding to inhibitory receptors, such as killer cell immunoglobulin-like receptors (KIRs), thereby dampening NK-cell activation. Accordingly, when MHC-I is downregulated, the inhibitory signals initiated by MHC-I are no longer present, leading to enhanced NK-cell activation and cytotoxicity [31]. Nonetheless, tumors have developed several mechanisms to escape NK-cell-mediated cytotoxicity. For example, tumors often produce factors, such as transforming growth factor β (TGF-β) and prostaglandin, that impair NK-cell function and block their infiltration into the tumor site [32]. Additionally, tumors may temporarily upregulate MHC-I expression in response to NK-cells, which allows them to avoid recognition by these cells [19,22,33]. As a result, tumors show plasticity in evading both NK- and T-cell-mediated cytotoxicity, thereby facilitating tumor immune escape. The importance of this plasticity is substantiated by the observation that colorectal cancer patients with reversible MHC-I downregulations have a worse prognosis compared to patients with irreversible MHC-I downregulations [22].

Dysregulated cell surface display of MHC-I complexes may be caused by genetic, epigenetic, transcriptional, and post-transcriptional alterations, which leads either to irreversible or reversible changes in MHC-I expression. Irreversible MHC-I defects are described in multiple types of cancer, including melanoma, head and neck squamous cell carcinoma (HNSCC), and lung, colorectal, bladder, laryngeal, and breast cancer [9,12,13,14,15,16,17] and arise due to structural genetic alterations, for example, in the class I heavy chain genes [12,13,14], β_2_M [9,15,16], and the TAP-encoding genes [17]. As MHC-I expression cannot be upregulated in tumors harboring this type of mutations, these irreversible defects will not be the main focus of this review. Reversible MHC-I dysregulations are characterized by the coordinated (post-)transcriptional downregulation of the HLA class I heavy chain, components of the APM, or β_2_M and have been observed in several types of tumors, including HNSCC, bladder cancer, and neuroblastoma [18,19,20]. The reversible nature of these pathway dysregulations makes them interesting targets in cancer immunotherapy.

## 3. MHC-I Expression Regulation

Three major transcription binding sites responsible for MHC-I heavy chain expression can be distinguished: an Enhancer A region, which can be recognized by NFkB; an interferon-stimulated response element (ISRE), which can be bound by interferon regulatory factor (IRF) 1; an SXY-module, which is recognized by NOD-like receptor family CARD domain containing 5 (NLRC5) [34]. Other APM genes are induced by the same set of transcription factors, resulting in the observation that the downregulation of APM players often coincides [35]. Accordingly, different pathways can lead to transcriptional activation of the MHC-I heavy chain-encoding genes as well as other genes responsible for the APM. In addition, these pathways can also act synergistically, magnifying the effect on MHC-I expression. By studying how MHC-I expression is normally regulated, we may be able to better understand the underlying mechanisms of MHC-I downregulation in tumors and find potential targets for cancer immunotherapy in order to reverse this downregulation.

## 4. Inducing MHC-I Expression in Cancer via NFkB Stabilization

The NFkB family consists of many inducible transcription factors, including NFkB1 (p50), NFkB2 (p52), RelA, RelB, and c-REL. During homeostasis, these proteins are sequestered in inactive cytosolic complexes by interaction with inhibitor of kB (IkB) family members (e.g., IkBα and p105). Upon pathway activation, IkB proteins are degraded, causing the release of NFkB transcription factors, allowing them to migrate to the nucleus to affect expression of target genes. Two major NFkB-inducing pathways can be distinguished: the canonical and the non-canonical NFkB pathway (Figure 2A,B) [36]. An elaborate description of pathway players important in this review can be found in Box 1. Although NFkB is constitutively active in most cancers [37], some tumors are able to downregulate NFkB signaling, which has been shown to impair MHC-I expression [35]. NFkB expression was found to be associated with favorable response to CPI therapy in patients with melanoma, indicating the potential of upregulating its expression to increase anti-tumor immunity [4,38]. To date, several regulators of the NFkB pathways have been described, revealing potential targets to enforce upregulation of MHC-I expression in cancer.

Box 1NFkB-mediated upregulation of MHC-I.The canonical NFkB pathway can be activated by stimulation of several immune receptors, including the TNFα Receptor (TNFR), Toll-like receptors (TLR), and cytokines receptors, by their ligands (e.g., TNFα, IL-1, or LPS). Upon receptor stimulation, TRAF2/6 and receptor-interacting protein 1 (RIP-1) are recruited, after which TRAF2/6 polyubiquitinates itself and RIP-1. In addition, TGF-β-activated kinase 1 (TAK1) is ubiquitinated and activated. TAK1 in turn is able to phosphorylate and activate the IkB kinase (IKK) complex (consisting of IKKα, IKKβ, and NF-kappa-B essential modulator (NEMO) (also known as IKKγ)). IKK needs to be recruited to the stimulated receptor before it can become phosphorylated, which is triggered by NEMO-mediated interaction of IKK with polyubiquitinated RIP-1 [39]. Subsequently, IKK phosphorylates IkB proteins, thereby targeting it for ubiquitination and degradation. Consequently, RelA-p50 and c-REL-p50 are released from their inactivating complexes, which results in translocation to the nucleus. Here they can activate transcription of their target genes, including MHC-I heavy chain- and other APM-encoding genes, by binding to kB enhancers in their promotors. A schematic overview of the canonical NFkB pathway can be found in Figure 2A [36].Activation of the non-canonical NFkB pathway is initiated by the binding of a ligand (e.g., TNFα, CD40L, or B-cell activating factor (BAFF)) to one of the TNFR superfamily members. NFkB-inducing kinase (NIK) becomes activated, which in turn is able to phosphorylate and activate IKKα, one of the subunits of the IKK complex. IKKα subsequently phosphorylates the carboxy-terminal serine residues of p100, which is an IkB-like molecule that sequesters the NFkB transcription factor RelB in the cytosol, resulting in ubiquitination and degradation of the C-terminus of p100. As a result, p52 is generated, which forms a heterodimer with RelB, thereby allowing migration to the nucleus, resulting in induction of transcription of target genes. A schematic overview of the non-canonical NFkB pathway can be found in Figure 2B [36].

### 4.1. Positive Regulators of NFkB Expression

TNFα forcefully stimulates NFkB signaling and subsequent MHC-I expression. However, its use in cancer therapy is limited due to severe toxicities [40]. Alternatively, retinoids have been described to induce MHC-I upregulation in cancer, which is suggested to be the result of stimulation of NFkB signaling, even though the exact mechanism is still under debate. Studies in neuroblastoma and embryonic carcinoma cells revealed that retinoids increase MHC-I expression via increased expression of NFkB p50 and RelA [41,42]. In addition, Vertuani et al. [43] reported that upregulation of proteasome subunits (Latent Membrane Protein (LMP)-2, -7, and -10) and increased half-life of MHC-I complexes are responsible for increased MHC-I expression in neuroblastoma cell lines. Retinoids are currently being used to treat several types of cancer, including neuroblastoma and promyelocytic leukemia, and are known to induce differentiation, apoptosis, and inhibition of proliferation of tumor cells [44].

Other compounds reported to positively affect NFkB signaling of which to date the effect on MHC-I expression has not been assessed, are betulinic acid, and calcium/calcineurin combined with protein kinase C (PKC) antagonists. Betulinic acid upregulates canonical pathway activity via boosting the activity of IKK as well as phosphorylation and degradation of IkBα in multiple cancer cell lines, including neuroblastoma, glioblastoma, and melanoma [45]. In addition, it induces apoptosis, inhibits topoisomerase I activity, and suppresses angiogenesis in cancer [46,47,48,49,50]. Secondly, a study into latent HIV infections in CD4+ T-cells revealed that combining calcium/calcineurin and the protein kinase C (PKC) antagonist prostatin causes synergistic activation of NFkB through a similar mechanism as betulinic acid [51]. A third strategy would be to increase expression of the E3 ubiquitin ligase Nedd4 [52], which triggers polyubiquitination and proteasomal degradation of N4BP1 [53], a suppressor of NFkB. In addition, in B-cells, Nedd4 has been reported to induce ubiquitination and degradation of TNF receptor-associated factor (TRAF) 3 via the TNFR CD40, thereby inducing activation of both the canonical and non-canonical NFkB pathway [52]. However, to date, no Nedd4 stimulatory compounds have been described.

Nonetheless, dysregulation of factors affecting downstream signaling of NFkB-inducing pathways makes it questionable whether NFkB activity, and thus MHC-I expression, can be restored by upstream pathway activation. As a result, therapeutic intervention to inhibit negative regulators of NFkB may be a more effective strategy to explore.

### 4.2. Negative Regulators of NFkB Expression

The pediatric tumor neuroblastoma is well known for its low MHC-I expression. We have previously identified two major negative regulators of MHC-I via NFkB signaling in neuroblastoma: Nedd4 Binding Protein 1 (N4BP1) and TNFα-induced protein 3 interacting protein 1 (TNIP1) [54]. TNIP1 affects canonical NFkB activation, whereas N4BP1 exerts an effect on both canonical and non-canonical pathway activation.

N4BP1 interacts with several proteins involved in (de)ubiquitination, including NEDD4, Cezanne-1, A20, and Itch [53,54] and potentially also binds to polyubiquitin itself [55,56]. Polyubiquitin binding proteins, like N4BP1, can compete with NEMO for polyubiquitin binding, thereby directly antagonizing activation of canonical NFkB [56]. Besides this, N4BP1 interacts with the deubiquitinase (DUB) Cezanne-1, which functions in the deubiquitination and stabilization of TRAF3 [54] and subsequently modulates the activation of both the canonical [54] and non-canonical NFkB [57] by affecting the degradation of NIK and IkB, and the induction of c-REL ubiquitination and proteasome-mediated degradation [58].

TNIP1 (also known as A20-binding inhibitor of NF-kB (ABIN)-1) is known for its stimulatory effect on the NFkB inhibiting DUB A20 and inhibition of TNF-induced apoptosis and is upregulated in several types of cancer [59]. Similar to N4BP1, TNIP1 inhibits NFkB by impairing NEMO-mediated translocation of IKK to the receptor site. The exact mechanism remains unclear, but it is suggested to be via competing with NEMO binding to RIP-1 as well as via binding to a polyubiquitin group on NEMO itself [60]. TNIP1 is also able to prevent processing the IkB p105 into p50 [61] and to interact with the NFkB inhibiting DUB A20, stimulating deubiquitination of NEMO, thereby impairing IKK activation and thus canonical NFkB activation [62]. IL-17 treatment has been shown to induce proteasome-dependent downregulation of TNIP1 [63]. This resulted in NFkB activation, thereby suggesting its potential to upregulate MHC-I expression. However, IL-17 has been shown to play a key role in the promotion of tumor progression by inducing chronic inflammation, tumor cell proliferation, angiogenesis, and metastasis, which may majorly limit the use of this cytokine to induce MHC-I expression in cancer [64].

Hence, targeting N4BP1 or TNIP1 will result in strong activation of NFkB signaling, which in turn boosts MHC-I expression. Indeed microRNA (miR) 28-5p is an inhibitor of N4BP1 and has been demonstrated to act as a tumor suppressor in many cancer types, including colorectal cancer, renal cell carcinoma, and hepatocellular carcinoma [65,66,67]. In addition, MiR-1180 and miR-486 have also been demonstrated to induce NFkB activation via targeting of several inhibiting players of the NFkB pathway, including Cezanne, A20, and TNIP1-3 [68,69]. Nevertheless, it should be taken into account that these miRs have also been associated with cancer cell growth, survival, migration, and progression, emphasizing the need to further investigate the effect of these miRs in cancer [70,71].

The NFkB inhibiting DUB A20 is also involved in TNIP-independent regulation of NFkB signaling [72]. A20 can form a ubiquitin-editing complex together with the ubiquitin binding protein TAX1 Binding Protein 1 (TAX1BP1) and the E3 ubiquitin-protein ligase Itch. This complex can deubiquitinate RIP-1 and TRAF6, thereby inhibiting TAK1 activation as well as NEMO-mediated recruitment and activation of IKK. In addition, A20 can induce K48-polyubiquitination of RIP-1, thereby targeting it for proteasomal degradation [73]. TAX1BP1 is responsible for Itch recruitment to A20, whereas Itch controls the interaction between A20 and its substrates RIP-1 and TRAF6, enabling inactivation of these substrates. Knockout of either of these proteins results in inadequate NFkB inhibition, indicating that all three proteins are indispensable in the functioning of this ubiquitin-editing complex [74]. Itch has been shown to be upregulated in several types of cancer, such as breast cancer and neuroblastoma, in which it was shown to play a major role in cancer progression [75,76,77,78,79]. The antidepressant clomipramine, its structural homologue norclomipramine, and 1,4-naphthoquinone 10E have been shown to reduce tumor growth and enhance chemotherapy in multiple cancer cell lines, including breast, prostate, and bladder cancer lines, as well as in a multiple myeloma xenograft model [80,81]. In addition, clomipramine has been shown to induce MHC-I expression in a rat model of experimental allergic neuritis [82]. Controversially, N4BP1 has also been described to negatively regulate Itch by blocking the binding of Itch to its substrates [53]. However, as both N4BP1 and Itch play an important role in suppressing NFkB signaling, the potential for therapeutic inhibition of Itch remains elusive.

Another DUB enzyme, cylindromatosis (CYLD) plays a role in the inhibition of NFkB signaling through the removal of polyubiquitin motifs from NEMO and another important upstream protein in the NFkB signaling pathway called TRAF2 [83]. Several miRs have been reported to target CYLD, including miR-1288, -196, and -372-5p [84,85,86]. Recently, as reviewed by Farshi and colleagues, several DUB inhibitors have been developed for the treatment of cancer, all targeting different DUB enzymes that play distinct roles in the promotion of cancer [87]. To date, no specific DUB inhibitors have been described to specifically target Cezanne-1, A20, or CYLD. Nonetheless, aspecific DUB small molecule inhibitors, such as ubiquitin aldehyde (UbaI) have been described, which may suppress the activity of these NFkB targeting DUBs [87]. However, the use of aspecific inhibitors is limited due to the severe toxicity and simultaneous targeting of beneficial DUBs. Therefore, additional research should be conducted to develop specific DUB inhibitors. A patent (https://patentscope.wipo.int/search/en/detail.jsf?docId=WO2017109488, WO2017109488) describes cyanopyrrolidine derivates as specific inhibitors of Cezanne-1. In addition, as Cezanne-1 and A20 show sequence homology, it might potentially be possible to find cross-reactive DUB inhibitors to boost NFkB activity. However, in line with the above described miRs, Cezanne-1, A20, and CYLD are described in tumor suppressive as well as tumor progressive processes, which may be a counterindication for the use of inhibitors of these proteins in cancer [88,89].

Altogether, these studies show that there are multiple proteins that play a role in suppressing NFkB signaling, which may subsequently lead to hampered MHC-I expression. Several therapeutic interventions have been described, which could potentially trigger NFkB expression in cancer. A major contradiction we should be aware of is that the NFkB pathway is constitutionally active in a variety of tumors, playing an important role in many tumor-promoting processes, including inflammation, invasion, proliferation, angiogenesis, and metastasis [37]. This is further substantiated by the dual effect of most described proteins [53], miRs [68,69,70,71], and therapeutic interventions [53,64,75,76,77,78,88,89] affecting NFkB pathway induction on tumor-suppressing and tumor-promoting processes. It has been hypothesized that NF-kB inhibits tumor growth in cancers with a low mutational burden (early stages of cancer and potentially pediatric tumors), but that accumulation of mutations may lead to a loss of tumor suppressive function and the oncogenic features of NF-kB can become more dominant [90]. Therefore, we should carefully study the effects of NFkB pathway induction in NFkB-downregulated tumors as this could shift the balance from immune evasion towards tumor progression, thereby potentially even hampering the efficacy of cancer immunotherapy.

## 5. Inducing MHC-I Expression in Cancer via Restored IFN Signaling

In addition to NFkB, Interferons (IFNs) play a significant role in the induction of MHC-I expression. During homeostasis, the signal transducer and activator of transcription (STAT) proteins are present in the cytosol in their inactive form. Upon pathway activation, STATs become phosphorylated and dimerize, allowing them to migrate to the nucleus to affect expression of target genes. Both type I and type II interferon pathways are able to induce dimerization of STATs, thereby upregulating MHC-I expression using different signaling pathways (Figure 3A,B). An elaborate description of pathway players important in this review can be found in Box 2. IFNs play an important role in the regulation of antigen processing and presentation and are described to exert pro- and anti-tumorigenic effects in various types of cancer as extensively reviewed by Musella et al. and Castro et al. [91,92]. Downregulation of both type I and II IFN-mediated pathways have been described as mechanisms involved in resistance to CPI- and adoptive cell therapy in melanoma and lung cancer, indicating the potential of interference in these pathways to increase anti-tumor immunity [4,5,6,10,93,94,95,96]. Several factors affecting IFN pathway expression have been described, revealing potential targets to modulate MHC-I expression in cancer.

Box 2Interferon-mediated upregulation of MHC-I.The type II IFN pathway can be activated by stimulation of the IFNG receptor (consisting of IFNGR1 and IFNGR2) by IFNγ, allowing binding, phosphorylation, and activation of Janus Activated Kinase (JAK) 1 and JAK2. The intracellular domain of IFNGR1 is phosphorylated, creating a docking site for signal transducer and activator of transcription (STAT)1, which is phosphorylated by JAK1/2, forms homodimers, and translocates to the nucleus to activate target gene expression via binding to gamma-activated site (GAS) elements in the promotor regions of IFN-stimulated genes (ISG). One of the genes induced by STAT1 signaling is the transcription factor IRF1, which in turn is able to bind to the ISRE site present in the MHC-I promoter [34]. A schematic overview of the type II IFN pathway can be found in Figure 3A [97].Activation of the type I IFN pathway is initiated by binding of type I IFNs (e.g., IFNα and IFNβ) to the IFNA receptor (consisting of IFNAR1 and IFNAR2), allowing binding, phosphorylation, and activation of JAK1 and tyrosine kinase 2 (TYK2). The cytoplasmic tail of IFNAR is phosphorylated, creating a docking site for STAT1-3, which are again phosphorylated by JAK1 to form homo and heterodimers. STAT1 homodimers can activate IRF1 expression as described for the type II IFN pathway. Additionally, STAT1/STAT2 forms a complex with IRF9 (called IFN-stimulated gene factor 3 (ISGF3), which is able to bind to the ISRE element in the MHC-I promotor. STAT3 homodimerization also occurs, however, this does not lead to MHC-I upregulation, but rather functions as a negative feedback loop to inhibit expression of pro-inflammatory genes. A schematic overview of the type I IFN pathway can be found in Figure 3B [98].

### 5.1. Positive Regulators of IFN Signalling

IFN signaling can be induced via treatment with IFN-inducing ligands, such as IFNα, IFNβ, and IFNγ. In addition, stimulation of several PRRs may result in downstream type I IFN production, thereby indirectly promoting MHC-I expression. IFNγ has been suggested to have the most potent effect on the expression levels of APM genes, including MHC-I, TAP, and ERAP, which may imply its beneficial effect in cancer immunotherapy [35,99,100]. However, in line with NFkB-inducing ligands, IFNs and PRR stimulation initiates a broad range of biological activities, thereby limiting its use in cancer therapy due to severe toxicities [101]. To avoid toxicities, targeted delivery of IFNγ to the tumor site may be of interest, for example by fusing it to an antibody specific for a tumor-associated antigen [102].

Another targeted approach would be a cellular NK-cell therapy strategy, as NK-cells are capable of either killing or upregulating MHC-I expression on MHC-I-lacking cells (via IFNγ secretion) [19,33,103]. The potential of NK-cell therapy is substantiated by observed correlations between the abundance of functional NK-cells and checkpoint inhibition efficacy in non-small cell lung cancer (NSCLC) [104,105]. However, several challenges remain, including anti-inflammatory responses within the tumor microenvironment (TME), which cause impaired NK-cell function and TME infiltration [19,33,106,107], as well as ex vivo expansion to generate sufficient NK-cell numbers, and in vivo persistence [108]. The suggested ability of tumors to both evade NK- and T-cell mediated cytotoxicity via plasticity in MHC-I expression is one of the anti-inflammatory mechanisms potentially decreasing efficacy of NK-cell therapy. Interestingly, several studies have observed a beneficial outcome for individuals with an inhibitory KIR genotype with lacking HLA-ligands (the ‘missing ligand’ genotype) undergoing cancer immunotherapy, as inhibitory KIRs without matching HLA-ligands cannot inhibit NK-cell cytotoxicity [109,110]. This led to the rationale of allogeneic, KIR genotype-mismatched NK-cell therapy strategies to optimize NK-cell cytotoxicity by decreasing HLA-dependent inhibition of NK-cells [111]. Besides this, it should be considered that the immune checkpoint programmed cell death-ligand 1 (PD-L1) is also shown to be upregulated by IFNγ [112,113]. As a result, IFNγ simultaneously induces MHC-I upregulation and T-cell suppression, thereby potentially creating a vicious circle of T-cell activation and inhibition.

### 5.2. Negative Regulators of IFN Signaling

The embryonic transcription factor double homeobox 4 (DUX4) has been shown to be upregulated in many cancer types, in which it causes downregulation of JAK1/2 and STAT1, thereby suppressing IFN target gene transcription, including MHC-I and other APM genes [114]. In line with this, DUX4 has been demonstrated to be significantly upregulated in non-responders to CPI, indicating that the DUX4-induced reduction in MHC-I expression results in decreased T-cell cytotoxicity in patients [114]. The exact mechanism in which DUX4 downregulates JAK1/2 and STAT1 remains to be elucidated. As DUX4 is studied in more detail in a disease called facioscapulohumeral dystrophy (FSHD), knowledge on treatment strategies could be gained from this disease. Bosnakovski et al. [115] reported that inhibition of p300, a histone acetyltransferase recruited by DUX4 to affect target gene expression, by a specific inhibitor, counteracted the effect DUX-4 overexpression in vitro and in vivo. Others reported before that p300 inhibition in multiple types of cancer led to suppressed proliferation [116]. However, whether p300 inhibition also restores IFN signaling remains to be elucidated. DUX4 downregulation was also observed when treated with p38 inhibitors in vitro and in vivo in FSHD models [117]. P38 inhibition has a favorable effect in cancer treatment, however, the wide variety of processes in which p38 is involved, including tumor suppressive processes, is a clear disadvantage of this inhibitor [118].

Lymphocyte adapter protein (LNK) has been reported to be able to negatively regulate IFN signaling via the induction of dephosphorylation of STAT1 [119,120]. LNK has been shown to be overexpressed in several solid tumors, including melanoma and ovarian cancer, and was found to be significantly increased in patients who did not respond to CPI therapy [120]. LNK has been reported to be targeted by miR-29b, miR-30-5p, miR-98, miR-181a-5p [121,122]. To date, no other therapeutic LNK inhibitors have been reported.

The peptidyl-prolyl isomerase Pin1 has been reported to induce ubiquitination and degradation of the type I IFN inducing transcription factor IRF3, which normally triggers a positive feedback loop upon type I IFN pathway stimulation [123]. Pin1 has been demonstrated to be elevated in multiple types of cancer, playing a role in stimulating several cancer-driving processes [124]. As Pin1 is involved in downregulating type I IFN signaling, its suppression may lead to enhanced MHC-I expression, thereby advancing T-cell-mediated cytotoxicity. Presently, several Pin1 inhibitors have been developed, such as miR-200b, miR-200c, and miR296-5p, the small molecules all-*trans* retinoic acid (ATRA) and KPT-6566, and the natural compound Juglone, which all showed anti-cancer activity in different types of cancer [125,126,127,128,129,130].

Several protein tyrosine phosphatases (PTPs) have been described to downregulate tyrosine phosphorylation in the JAK/STAT pathway, thereby interfering with induction of both type I and II IFN signaling. For example, PTPN1 dephosphorylates TYK2 and JAK2 [131], PTPN2 dephosphorylates JAK1, and PTPN11 (SHP2) has been demonstrated to inhibit phosphorylation of JAK1, STAT1, and STAT2 [132]. Tyrosine phosphatases have been shown to be elevated in multiple types of cancer, including breast, ovarian, gastric cancer, and glioma, in which they promote tumor growth, survival, and metastases [133]. To date, several efforts have been attempted in the development of tyrosine phosphatase inhibitors, including the PTPN11 inhibitor sodium stibogluconate, and the PTP1B inhibitor MSI-1436C, which are now being tested in clinical trials for various cancer types [133,134,135]. In addition, miR-155 has been reported to target several negative regulators of IFN signaling, including PTPN2, which has been shown to result in increased IFNγ production by T-cells within the TME, thereby promoting MHC-I expression [136]. In contrast, miRNA-155 has been demonstrated to be overexpressed in multiple types of cancer, playing a pivotal role in oncogenesis, which may limit its use in cancer [137].

The ubiquitin ligases RING finger protein 2 (RNF2) and Smad ubiquitination regulatory factor-1 (Smurf1) and the protein inhibitor of activated STAT (PIAS) all function via inhibiting transcription activation of STAT1 [138,139,140]. These proteins have been shown to be overexpressed in several types of cancer, including melanoma, gastrointestinal tumors, lymphoma, and pancreatic cancer, thereby playing an active role in cancer promotion [141,142,143,144,145,146]. Likewise, the ubiquitin ligase DCST1 is able to induce ubiquitination and degradation of STAT2, which results in hampered type I IFN signaling and has also been shown to be elevated in various cancers [147]. The small molecule inhibitors PRT4165 and A01 successfully inhibit, respectively, RNF2 and Smurf1 in vitro [148,149]. In addition, indirect inhibition of RNF2 was reported via inhibition of MEK-induced activation by the MEK-inhibitor trametinib [143]. To date, no DCST1 inhibitors have been described.

Finally, the protein kinase D2 (PKD2) has been shown to dampen type I IFN signaling via stimulating ubiquitination and endocytosis of IFNAR1, thereby causing rapid turnover of the type I IFN receptor [150]. PKD2 has been associated with multiple types of cancer, functioning by promoting tumor progression and blocking type I IFN signaling [151]. To date, several inhibitors of PKD2 have been developed, for example the small molecule inhibitor CRT0066101 and the pyrazolopyrimidine pan-PKD inhibitor SD-208, which both show anti-tumor activity in xenograft mouse models of various cancers [152,153].

Taken together, several proteins have been implicated in impaired IFN signaling in cancer. Accordingly, therapeutic intervention of these dysregulations may be beneficial when combined with immunotherapy to increase MHC-I expression and enhance T-cell-mediated cytotoxicity. However, many of these proteins have not been associated with MHC-I downregulation before, highlighting the need to investigate their effect on MHC-I expression. In addition, as stated for NFkB, IFNs are also reported to exert both pro- and anti-tumorigenic effects in various types of cancer [91,92], indicating the need to closely study these effects.

## 6. NLRC5-Mediated Upregulation of MHC-I

Only recently, NLRC5 was discovered as a third key regulator of MHC-I transcription [154]. NLRC5 is expressed in response to IFN-mediated signaling, in particular via IFNγ, by binding of a STAT1 homodimer to GAS in the NLRC5 promoter [154,155]. Interestingly, this STAT1 binding site in the NLRC5 promoter was shown to partially overlap with an NFkB binding site, suggesting that NLRC5 expression may be induced by NFkB as well [156]. Moreover, it has been suggested that the NLRC5 promoter contains an ISRE site, which may allow transcriptional activation by IRF1. Upon its expression, NLRC5 migrates to the nucleus, where it assembles with several proteins, including RFX, ATF1/CREB, and the NFY complex, resulting in the formation of the MHC enhanceosome. This complex is able to bind the SXY module in the MHC-I promoter, causing the induction of MHC-I expression. Additionally, NLRC5 is able to recruit transcriptional initiation and elongation factors, and histon-modifying enzymes, for example histon acetyltransferases and methyltransferases, which may further enhance transcriptional activation of MHC-I [82,157]. (Epigenetic) downregulation of NLRC5 expression has been observed as a mechanism of immune evasion in several types of cancer, including colorectal, ovarian, breast and uterine cancers [158,159]. NLRC5 has been reported to elicit anti-tumor immunity by enhancing antigen processing and presentation in melanoma [160]. In addition, NLRC5 expression correlates with response to CPI in melanoma (https://pubchem.ncbi.nlm.nih.gov/patent/US2017321285, US20170321285A1). Controversially, as also observed for NFkB- and IFN-mediated signaling pathways, NLRC5 has been reported to exert both pro- and anti-tumorigenic effects in various types of cancer (reviewed by Tang et al. [161]), indicating the need of careful evaluation of upregulation of NLRC5 in cancer. To date, no specific NLRC5 targeting compounds have been described. Nonetheless, as NLRC5 activation pathways largely overlap with the above described pathways, compounds inducing these pathways may also result in NLRC5 activation.

## 7. Inducing MHC-I Expression in Cancer via STAT3 Inhibition

As briefly touched upon in the section above, STAT3 becomes activated upon type I IFN signaling to function as a negative feedback loop to inhibit expression of pro-inflammatory genes, thereby contributing to balanced immune responses [98]. STAT3 is implicated in dampening a variety of immune responses, including inhibition of both NFkB- and IFN-mediated pathways. STAT3 is able to inhibit activation of the IKK-complex, thereby preventing phosphorylation and degradation of IkB, which results in sequestering of NFkB transcription factors in the cytosol.

STAT3 has been shown to play a key role in the promotion of cancer by mediating proliferation, survival, invasion, and metastasis [162]. STAT3 inhibition resulted in increased anti-tumor activity and superior responses to immunotherapy and immunogenic chemotherapy in several pre-clinical studies and trials [163,164,165,166,167,168,169,170,171,172]. Combinations of CPI and STAT3 inhibitors are currently tested in clinical trials [173]. STAT3 can be inhibited via upstream JAK inhibition as well as by direct inhibition of STAT3. Several specific STAT3 inhibitors have been described, including JSI-124 (cucurbitacin I) [163], static [166], indirubin [164], resveratrol [165], and the antibiotic nifuroxazide [170]. In addition, the multitarget tyrosine kinase inhibitors sorafenib and sunitinib are also reported to specifically target STAT3 phosphorylation and activation [167,168].

As STAT3 overexpression inhibits both NFkB- and IFN-signaling pathways, specific inhibition of STAT3 may be an interesting strategy to target downregulation of these pathways and potentially induce MHC-I upregulation. However, as is the case for other NFkB- and IFN-signaling pathway inhibitors, these pathway inhibitors also exhibit favorable effects, as for example indicated by the fact that STAT3 also elicits tumor-suppressive functions [174].

## 8. Inducing MHC-I Expression in Cancer via STING Induction

Stimulator of Interferon Genes (STING) is a DNA-sensing molecule which activates upon encountering foreign DNA by cyclic dinucleotide recognition [175]. Upon cyclic dinucleotide recognition, STING becomes activated and forms a complex with TANK-binding kinase 1 (TBK1), which in turn is able to phosphorylate and activate both RelA and IRF3, thereby stimulating both NFkB- and type I IFN-activation [176]. To date, several STING agonists have been developed to exploit this response in cancer. One example is the STING agonist SB 11285, which is currently tested in clinical trials in several solid tumors, including HNSCC and melanoma (ClinicalTrials.gov Identifier: NCT04096638), and previously showed strong anti-tumor immunity induction in mouse models [177].

As STING induced both NFkB- and IFN-pathway activation, agonizing its activity may be an interesting strategy to upregulate these pathways to potentially induce MHC-I upregulation. This is further substantiated by the correlation between STING expression and HLA-associated genes in neuroblastoma [178]. In addition, preclinical studies have reported superior effects of CPI therapy in combination with STING agonists [179,180]. Combining CPI with STING agonists is currently being tested in clinical trials [181]. The multi-facet involvement of STING in immune activation, however, underlines the need to study its effect to improve therapy outcomes while not compromising treatment safety.

## 9. Well-Known Oncogenic Pathways Affect MHC-I Expression

Various oncogenic pathways have been reported to affect expression of MHC-I, β_2_M, and other APM components in cancer, including the MAPK-, epidermal growth factor receptor (EGFR), HER2, c-MYC, and n-MYC pathway [182,183,184,185,186,187]. MAPK pathway activation is suggested to negatively influence MHC-I expression via decreased IRF1 activity and STAT1 expression [188]. The MEK inhibitors trametinib and cobimetinib have been demonstrated to enhance IRF1 expression and increased STAT1 phosphorylation in human keratinocytes [188]. Watanabe et al. [189] reported that treatment of a NSCLC cell line with trametinib also increased MHC-I expression in vitro. Another MEK inhibitor, selumetinib, increased MHC-I expression in papillary thyroid cancer cell lines [2]. In addition, upstream inhibition of BRAF by vemurafenib and dabrafenib also induced MHC-I and β_2_M upregulation in multiple melanoma cell lines [190].

Overexpression of the EGFR family member HER2/neu was shown to be inversely correlated with MHC-I expression in different types of cancer, including breast cancer, esophageal squamous cell carcinoma, and melanoma [191,192]. Additionally, overexpression of EGFR was associated with less potent responses to cancer immunotherapy in NSCLC and neuroblastoma [193,194]. EGFR activates PTNP11, which was mentioned above as a STAT1 inactivator [195]. PTNP11 also inhibits RAS GTPase-activating protein, which normally functions in downregulating MAPK signaling, thereby also contributing to impaired IFN signaling via the MAPK pathway [196]. A third mechanism in which EGFR signaling impairs MHC-I expression is the promotion of STAT3 activation [197]. HER2 in turn also induces proteasomal degradation of the tumor suppressor Fhit, which normally upregulates expression of MHC-I, APM components, and β_2_M via an as-yet unclear mechanism [142]. Hence, targeting tyrosine kinase receptors, for example by the anti-EGFR antibodies nimotuzumab and cetuximab, and the EGFR inhibitors afatinib, erlotinib, and gefitinib, results in enhanced expression of MHC-I and APM components in different cancer cell lines as well as in cancer patients [185,198,199,200,201,202].

Finally, oncogenes c- and n-MYC have both been associated with MHC-I downregulation [186,187,203]. N-MYC has been shown to decrease MHC-I expression in rat neuroblastoma cell lines through the inhibition of the NFkB transcription factor p50 [203]. On the contrary, Forloni et al. [204] showed that that the downregulation of MHC-I in human neuroblastoma cells was not caused by n-MYC. In line with this, we did not identify n-MYC as a negative regulator of NFkB [54]. However, one limitation of our study was the use of an early phase CRISPR/Cas 9 library, which impairs the ability to fully confirm the results found in previous studies. Recently, Yang and colleagues showed that c-MYC expression induction by the Wnt/B-catenin pathway is responsible for MHC-I downregulation in glioma [187]. This again hints towards involvement of MYC-family members in MHC-I expression regulation in cancer.

Altogether, by targeting these oncogenic pathways, we potentially will not only be able to impair tumor growth and proliferation, but also exhibit anti-tumor responses via increasing the immunogenicity of the tumor.

## 10. Chemotherapy- and Radiation-Induced MHC-I Expression

Certain chemotherapeutic therapy regimens are known to induce anti-tumor immune responses without inducing classical immunogenic cell death (ICD). Hodge and colleagues showed that both docetaxel-induced ICD-sensitive and resistant tumor cells expressed increased levels of APM components, including TAP2, calnexin and calreticulin [205]. In addition, immunochemotherapy combining IFNα and 5-fluorouracil treatment resulted in increased MHC-I expression via STAT1/2 activation in murine pancreatic cancer models [206]. Besides this, several topoisomerase inhibitors (e.g., topotecan, irinotecan, and etoposide), microtubule stabilizers (e.g., paclitaxel and vinblastine), cisplatin, and ionizing radiation elevate MHC-I surface expression, which is thought to be induced via NFkB stabilization and IFNβ secretion [207,208,209]. These studies indicate that cytostatic drugs may, besides direct induction of cell death, also be involved in decreasing tumor immune escape via upregulation of antigen presentation, thereby making these drugs interesting candidates for combination therapy to improve immunotherapeutic strategies in cancer.

## 11. Epigenetic Silencing Affecting MHC-I Expression

Another common reversible defect in MHC-I antigen presentation is the occurrence of epigenetic modulation, which may impair the transcription of MHC-I, APM components, β_2_M, or MHC-I regulatory proteins. Histon deacetylation (HDAC) has been reported to reduce expression of MHC-I and key components of the APM, such as the proteasome subunits LMP-2 and LMP-7, and TAP in multiple types of cancer, including neuroblastoma, glioma, Merkel cell carcinoma, cervical cancer, and melanoma [187,210,211,212]. HDAC inhibitors, of which several already are FDA approved (e.g., Romidepsin, Vorinostat and Panobinostat), show increased expression of MHC-I and APM components both in vitro and in vivo [187,210,211,213]. Multiple other HDAC inhibitors are currently tested in clinical trials either alone or in combination with other drugs in various types of cancer, including melanoma, breast cancer, and lung cancer [214].

Epigenetic modulation by DNA hypermethylation has also been reported to cause reduced expression of MHC-I and related genes, which may be reversed by treatment with DNA methyltransferase inhibitors (DNMTi). The DNMTis guadecitabine, 5-azacytidine, and decitabine increased MHC-I expression in breast cancer, melanoma, and neuroblastoma cell lines [54,215,216,217]. Interestingly, combinatory treatment with TNFα, IFNγ, or knockout of known inhibitors of NFkB signaling (TNIP and N4BP1) further exaggerated the effect of these epigenetic modulators [54,215,217]. Additionally, increased MHC-I expression was observed in breast cancer patients treated with a combination of DNMT and HDAC inhibitors [216], indicating the potential of epigenetic modification to reintroduce MHC-I expression in tumors.

## 12. Discussion

MHC-I-mediated antigen presentation is crucial for CD8^+^ T-cell cytotoxicity and is one of the key factors in endogenous adaptive immune response development as well as T-cell mediated immunotherapy efficacy in cancer. The often intrinsically reversible nature of these dysregulations provides an opportunity to restore MHC-I expression and therewith adaptive anti-tumor immunity. In this review, we highlight that MHC-I-mediated antigen presentation is a complex, multi-faceted process, which can be dysregulated at many levels. As APM players are often induced by the same set of transcription factors, downregulation of APM players often coincides. To identify therapeutic targets to increase immunotherapy efficiency in cancer, a better understanding of the underlying mechanisms of MHC-I downregulation in tumors is required. A summary of regulators of MHC-I expression and potential therapeutic strategies described in this review can be found in Table 1 and Figure 4.

Dysregulation of factors affecting the downstream signaling of MHC-I-inducing pathways makes it questionable whether pathway activity can be restored by upstream pathway activation. As a result, therapeutic intervention to inhibit negative regulators of these pathways, possibly combined with stimulation of positive regulators of these pathways, may be the most promising strategy to explore.

Downregulation of MHC-I is an important factor contributing to the immune evasion of tumors. However, as underlined by the observed resistance of solid tumors to CAR-therapy as well as by IFNγ-mediated upregulation of PD-L1 [112,113], immune evasion of tumors is a multifaced process. As a result, the authors hypothesize that combination therapy targeting several tumor-immunomodulatory processes simultaneously will be necessary to avoid immune evasion of tumors. Several combinations of immunomodulatory drugs, including combining CPI with STAT3 inhibitors or STING agonists, show promising results in (pre-)clinical studies [171,172,173,179,180,181].

An important contradiction in enhancing NFkB-, IFN-, and NLRC5-mediated MHC-I expression is the dual role of these pathways in cancer [37,91,92,161]. They have been described in both tumor-promoting as well as tumor-suppressive mechanisms in several types of cancer. The authors hypothesize that, depending on the activation status of these pathways, its function will either be to promote or suppress tumor progression. Hence, it is important to study the effect of (pharmacological) enhancement of these pathways in MHC-I lacking tumors, as overactivation might shift the balance from immune evasion towards tumor progression. However, the observation that several NFkB/IFN upregulatory drugs do show beneficial effects in the treatment of several types of cancer highlights the potential of these drugs in improving immunotherapy outcomes in cancer.

## 13. Conclusions

In conclusion, MHC-I antigen presentation is a complex process regulated by multiple pathways that can be pharmacologically targeted on multiple levels to increase pathway activation and trigger MHC-I expression in cancer. Increasing antigen presentation in MHC-I-downregulated tumors is key to increase adaptive anti-tumor immunity and improve (immuno)therapy efficacy.

## Figures and Tables

**Figure 1 cancers-12-01760-f001:**
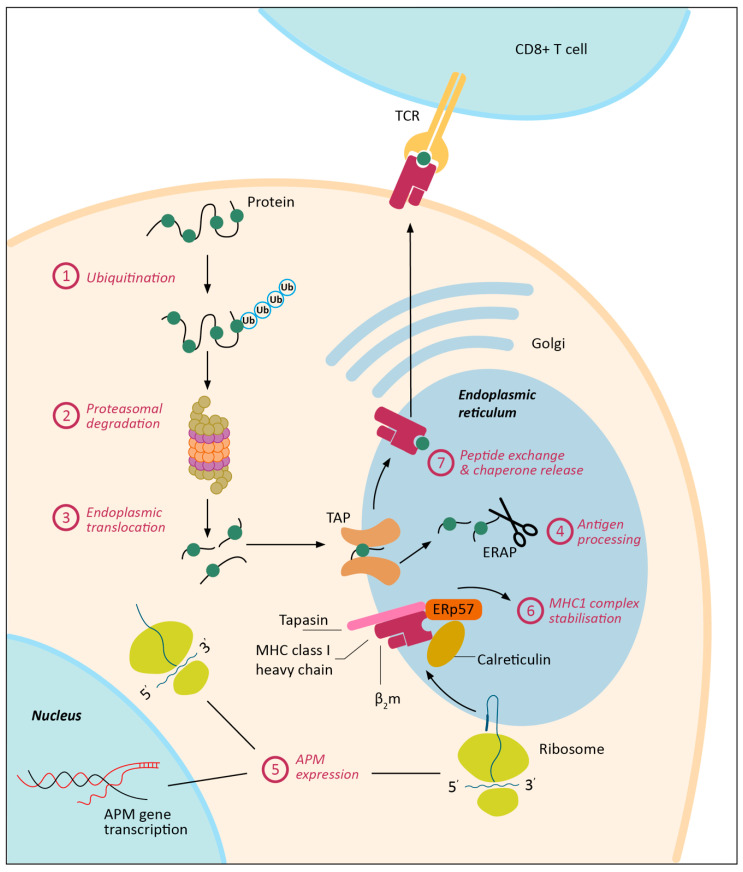
Major histocompatibility complex class I (MHC-I) antigen processing and presentation is a complex, multi-step process and can be dysregulated in cancer at multiple levels. APM = antigen processing machinery; ERAP = ER aminopeptidases; MHC-I = major histocompatibility complex I; TCR = T-cell receptor

**Figure 2 cancers-12-01760-f002:**
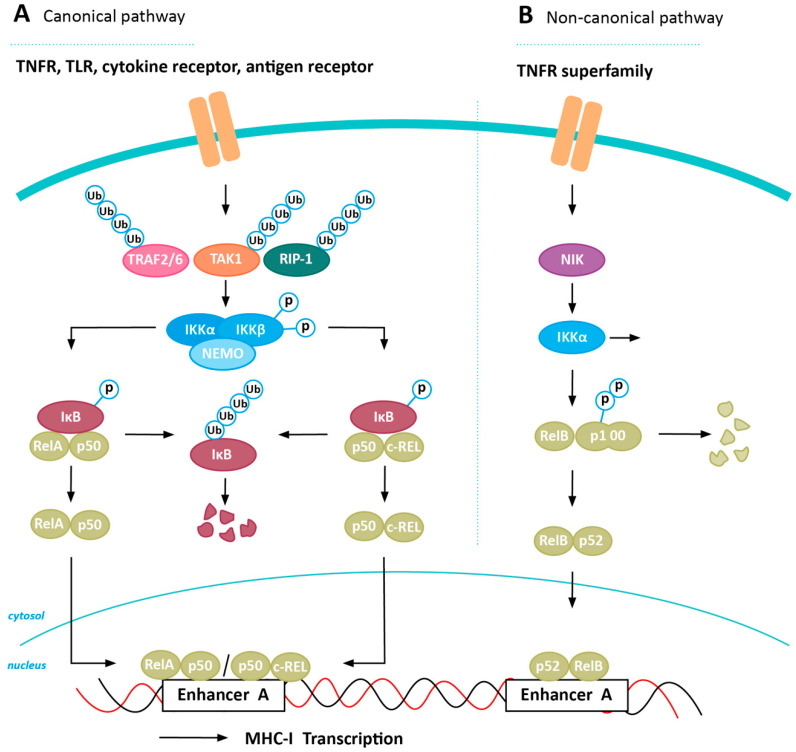
NFkB-induced expression of MHC-I. Transcriptional activation of MHC-I heavy chain, but also other genes encoding for antigen processing machinery (APM) proteins, can be initiated by both the canonical (**A**) and non-canonical (**B**) NFkB pathway. IkB = inhibitors of kB; IKK = IkB kinase; NEMO = NF-kappa-B essential modulator; NIK = NFkB-inducing kinase; RIP-1 = receptor-interacting protein 1; TAK1 = TGF-β-activated kinase 1; TLR = toll-like receptor; TNFR = TNFα receptor; TRAF = TNF receptor-associated factor.

**Figure 3 cancers-12-01760-f003:**
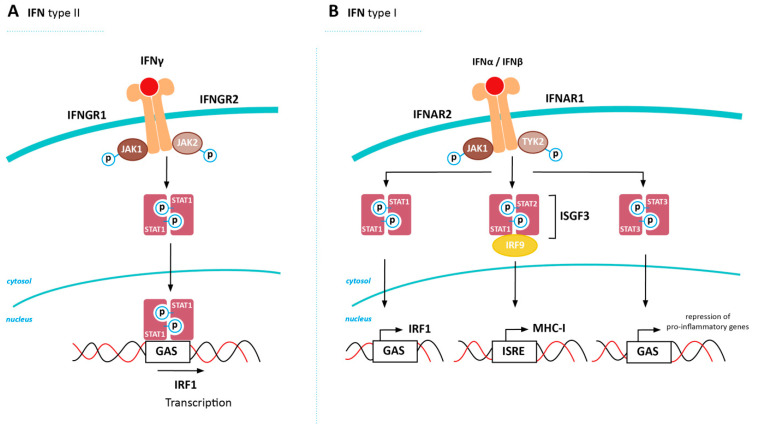
Interferon (IFN)-induced expression of MHC-I. Transcriptional activation of MHC-I heavy chain, but also other genes encoding for antigen processing machinery (APM) proteins, can be initiated by both the type II (**A**) and I (**B**) IFN pathway. GAS = gamma-activated site; IFNAR = IFNα receptor; IFNGR = IFNγ receptor; ISGF3 = IFN-stimulated gene factor 3; IRF = interferon regulatory factor; ISRE = interferon-stimulated response element; JAK = Janus Activated Kinase; STAT = signal transducer and activator of transcription; TYK2 = tyrosine kinase 2.

**Figure 4 cancers-12-01760-f004:**
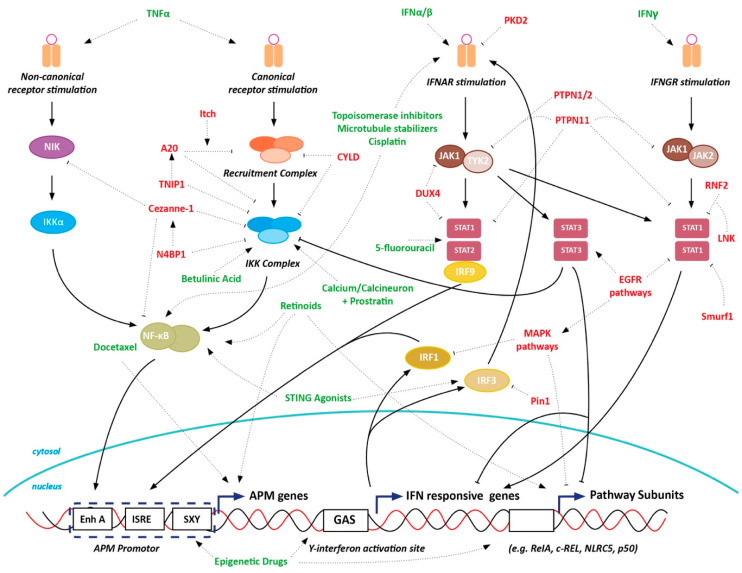
Potential therapeutic interference to boost MHC-I antigen presentation of tumors. Targetable negative pathway regulators are shown in red, (general groups of) compounds positively affecting pathway activation are shown in green.

**Table 1 cancers-12-01760-t001:** Overview of described pathways and potential therapeutic strategies to boost MHC-I antigen presentation.

Regulation	Pathway	Effector	Described Mechanism(s)	Potential Therapeutic Strategy	Clinical Status
Negative Regulators	NFkB pathway	N4BP1	1. Cezanne-1 stabilization 2. Preventing recruitment of NEMO to RIP1	miRNA-28-5p [65,66,67,70,71]	Pre-clinical
		Cezanne-1	Deubiquitination and stabilization of TRAF3	Aspecific DUB inhibitors (e.g., Ubal) [87] miR-1180 [69], miR-486 [68] Cyanopyrrolidine derivates (WO2017109488)	Pre-clinical
		TNIP1	1. Preventing recruitment of NEMO to RIP1 2. Preventing degradation of the IkB p105 into p50 3. A20 stabilization	IL-17 [63] miR-1180 [69], miR-486 [68]	Pre-clinical
		A20	1. Deubiquitination of NEMO 2. Deubiquitination of RIP-1 and TRAF6 3. Degradation of RIP-1	Aspecific DUB inhibitors (e.g., Ubal) [87] miR-1180 [69], miR-486 [68]	Pre-clinical
		TAX1BP1	Itch recruitment to A20		
		Itch	Controlling interaction between A20 and RIP1/TRAF6	Clomipramine and norclomipramine [80] 1,4-naphthoquinone 10E [81]	FDA-approved antidepressants Pre-clinical
		CYLD	1. Deubiquitination of NEMO 2. Deubiquitination of TRAF2	miR-1288 [84], miR-186 [85], miR-362-5p [86]	Pre-clinical
	Type I/II IFN pathway	DUX4	1. JAK 1/2 downregulation 2. STAT2 downregulation	p300 inhibitors [115] p38 inhibitors [117]	Pre-clinical Several Phase II Trials (Rheumatoid Arthritis, Asthma, LMNA-related cardiomyopathy)
		LNK (SH2B3)	Dephosphorylation of STAT1	miR-29b [122], miR-30-5p [121], miR-98 [121], miR181a-5p [121]	
		PTPN1 (PTP1B)	1. Dephosphorylation of TYK2 2. Dephosphorylation of JAK2	MSI-1436C [135]	Phase I Trial, metastatic breast cancer
		PTPN2 (TC-PTP)	1. Dephosphorylation of JAK1	miRNA-155 [136]	Pre-clinical
		PTPN11 (SHP2)	1. Dephosphorylation of JAK1 2. Dephosphorylation of STAT1 3. Dephosphorylation of STAT2	Sodium stibogluconate [134]	Phase II Trial, Leishmaniasis, stage IV melanoma, advanced solid tumor
		RNF2	Polyubiquitination of STAT1, resulting in release from DNA	Trametinib [143] PRT4165 [148]	FDA-approved in several advanced-stage cancers Pre-clinical
		Smurf1	Degradation of STAT1	A01 [149]	Pre-clinical
		PIAS	Inhibiting STAT1 promotor recruitment		
	Type I IFN pathway	DCST1	Ubiquitination and degradation of STAT2		
		PKD2	Stimulating ubiquitination and endocytosis of IFNAR1	CRT0066101 [152] SD-208 [153]	Pre-clinical
		Pin1	Ubiquitination and degradation of IRF3	miR-200b [126], miR-200c [127], miR-296-5p [128] ATRA [129] KPT-6566 [130] Juglone [125]	Pre-clinical FDA-approved in cancer Pre-clinical Pre-clinical
	NFkB and type I/II IFN pathway	STAT3	1. Negative feedback loop of pro-inflammatory type I signaling 2. Inhibition of IKK activation	STAT3 inhibitors: JSI-124 [169] Indirubin [164] Resveratrol [165] Nifuroxazide [170] Static [166] Sorafenib [167] Sunitinib [168]	Pre-clinical Phase 3/4 trials in acute promyelocytic leukemia, phase 2/3 in dermatitis and psoriasis Phase 2/3 in congestive heart failure, Friedreich Ataxia, Gulf War Illness, Lymphangioleiomyomatosis, and infertility FDA-approved antibiotic Pre-clinical Pre-clinical Pre-clinical
		EGFR receptor family (including HER2/neu)	1. STAT1 inactivation 2. Stimulation of MAPK pathway 3. STAT3 activation 4. Fhit degradation	Tyrosine Kinase inhibitors: EGFR inhibitors (e.g., nimotuzumab [198], cetuximab [199], afatinib [185], erlotinib [185], gefitinib [202]).	FDA-approved in cancer
		NMYC	Inhibition of p50 (although inconsistent)		
		MAPK pathway	1. Decreases IRF1 activity 2. Decreases STAT1 expression	Tyrosine Kinase inhibitors: MEK-inhibitors (e.g., trametinib [188,189], cobimetinib [188], selumetinib [2]) BRAF inhibitors (e.g., vemurafenib [184,190], dabrafenib [41])	FDA-approved in cancer, selumetinib in phase 3 FDA-approved in cancer
	NFkB pathway	NFkB-inducing receptor stimulation	1. TNFR superfamily stimulation 2. PRR Receptor stimulation 3. IL-1 receptor stimulation	TNFα [40]	Phase I-III trials in cancer, all localized infusions
		IKK/IkBα	1. Boosting IKK-activity 2. Degradation of IkBα	Betulinic acid [45]	Phase I trial, Anxiety
		IKK	Boosting IKK-activity	Calcium/calcineurin + prostratin [51]	Pre-clinical
		Nedd4	1. Degradation of N4BP1 2. Degradation of TRAF3		
		p50, RelA, and LMP-2/-7/-10	1. Enhanced expression of p50 2. Enhanced expression of RelA 3. Enhances expression of LMP-2/-7/-10	Retinoids [41,218]	FDA-approved in several diseases, including cancers
	Type I IFN pathway	IFN-inducing receptor stimulation	1. PRR Receptors stimulation (e.g., RNF135, TRIM25, ISG15, NAB2) 2. IFNA Receptor stimulation	IFNα [100,101] IFNβ therapy [101]	FDA-approved in hepatitis B & high-risk melanoma Phase III trial in relapsing multiple sclerosis, Phase I in refractory solid tumors
	Type II IFN pathway	IFN-inducing receptor stimulation	IFNG Receptor stimulation	IFNγ-1b NK cell therapy [19,33,103,106,107,219]	FDA approved as localized injection in chronic granulomatous disease (CGD) and severe, malignant ostepetrosis (SMO), phase I-III in cancers Phase I/II Trial in several cancer types
	NFkB- and type I IFN pathway	STING	1. Phosphorylation and activation of RelA 2. Phosphorylation and activation of IRF3	STING agonists (e.g., SB 11285 [177])	Phase I Trial in patients with advanced solid tumors
	NFkB- and type I IFN pathway		1. Increasing APM expression (Calreticulin, TAP2, calnexin) 2. Enhancing type I IFN signaling 3. NFkB stabilization 4. IFNβ secretion	Docetaxel [205] 5-Fluorouracil [206] Topoisomerase inhibitors (e.g., topotecan [207,209], irinotecan [208], and etoposide [207] Microtubule stabilizers (e.g., paclitaxel [207,209] and vinblastine [207]) Cisplatin [207]	All FDA-approved in cancer
Epigenetic modification	NFkB and type I/II IFN pathway		1. Histon acetylation to decrease genome accessibility 2. DNA methylation to decrease genome accessibility	HDAC inhibitors: (e.g., Romidepsin, Vorinostat [212] and Panobinostat) [1,211,212,213,214] DNMTis (e.g., 5-azacytidine [212], Decitabine [54], and Guadecitabine [216])	All FDA-approved in cancer FDA approved, Guadecitabine phase II trials in cancer).

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
