# Peer review of "MHC Class I Downregulation in Cancer: Underlying Mechanisms and Potential Targets for Cancer Immunotherapy"

_cancers, 2020, doi:10.3390/cancers12071760_

Round 1

Reviewer 1 Report

The manuscript by Cornel A.M et al. is nicely written and present providing a detailed overview of the mechanisms and potential targets for cancer immunotherapy.

In my opinion, the manuscript can be accepted in the present form.

Author Response

No comments to answer.

Reviewer 2 Report

General comments:

This review is very interesting, well written and relevant for cancer immunology and immunotherapy. It summarizes many different mechanisms of the regulation of tumor MHC class I expression in different types of cancer based on the analysis of more than 200 publications. Importantly, it gives an overview of possible targets for therapeutic intervention.

It is well established that the loss or downregulation of MHC class I is one of the important mechanisms of cancer immune escape associated with the inhibition of tumor antigen presentation to cytotoxic T-cells.  It is also known that there are different molecular mechanisms that cause lack of MHC-I expression: genetic aberrations in the HLA and APM genes and reversible regulatory defects. In this manuscript the authors focus mostly on the regulatory defects causing MHC-I downregulation, which can be reversed by cytokines and transcriptional or epigenetic modulators.

Specific comments for minor revision:

  1. Although the regulatory mechanisms are more frequent, the observed resistance to cancer immunotherapy has been mostly linked to the genetic alterations that lead to irreversible loss of MHC-I expression. These defects cannot be reversed by cytokines or immunotherapy and lead to the outgrowth of tumor escape variants. It should be emphasized in the review.

As a matter of fact, the authors quote many key publications describing the high incidence of b2m mutations and loss of heterozygosity at chromosomes carrying HLA genes in patients not responding to IC and other types of cancer immunotherapy.  

  1. The following statement is not correct and does not agree with the corresponding citations: “The importance of MHC-I downregulation in immune evasion is substantiated by the observed inverse correlations between MHC-I expression on tumor cells and the amount of tumor infiltrating lymphocytes (TILs) in several cancers [2,3]”.

There is a lot of both clinical and experimental evidence indicating that MHC-I positive tumors are highly infiltrated with TILs. According to the reference 3,  Both MHC class I and β2-microglobulin expression was reduced or absent in 76% of PTC specimens and was associated with reduced tumor-infiltrating immune cells”.

Reviewer 3 Report

The manuscript submitted by Cornel et al. is a good review of processes related to MHC class I gene expression in cancer. The topic is important for cancer immunotherapy and as such it has been thoroughly studied over the last decades. The paper has the potential to help interested readers to get better oriented in this area.

The text is well structured and comprehensive, with a lot of details. However, it is descriptive rather than analytical. It is probably for this reason that sections Discussion and especially Conclusion are really short and not too informative.

The review merits to be published. I have some minor comments that could contribute to a better understanding of the importance of the topic reviewed.

Due to its focus on some detailed descriptions, the review did not cover all aspects of the problem, especially those related to immunotherapy mentioned in the title. More attention could be paid to immunity-related events occurring in tumors at the cellular level, i.e. to mechanisms of tumor escape from cytolytic immune activities. This would help readers to better understand how the question why some tumors are not responsive or develop resistance to therapy are addressed.

Little attention also has been paid to the importance of NK cells for future immunotherapy strategies. KIR/HLA class-I molecules were associated with clinical outcomes of cancer via several mechanisms, and inhibitory KIRs are considered as possible targets of immunotherapy strategy. I therefore recommend to better describe the function of MHC class I binding NK cell receptors (chapter 2, third paragraph, lines 74-79) and to discuss this aspect in Discussion.
